# Caries Experience in Preschoolers in Three Ecuadorian Communities

**DOI:** 10.3390/children10071123

**Published:** 2023-06-28

**Authors:** Eleonor María Vélez León, Alberto Albaladejo Martínez, Mónica Alejandra Preciado Sarmiento, María Augusta Cordero López, Ana del Carmen Armas, Liliana Soledad Encalada Verdugo, María Melo

**Affiliations:** 1Department of Surgery, Faculty of Medicine, University of Salamanca, 37007 Salamanca, Spain; albertoalbaladejo@usal.es; 2School of Dentistry, Catholic University of Cuenca, Cuenca 010107, Ecuador; mcorderol@ucacue.edu.ec (M.A.C.L.); lencalada@ucacue.edu.ec (L.S.E.V.); 3Ministry of Public Health, Cuenca 010107, Ecuador; monica.preciado@est.ucacue.edu.ec; 4School of Dentistry, Hemisferios University, Quito 170527, Ecuador; ana_del_ec@yahoo.es; 5Department of Stomatology, Faculty of Medicine and Dentistry, University of Valencia, 46010 Valencia, Spain; m.pilar.melo@uv.es

**Keywords:** caries, prevalence, childhood, children, ICDAS

## Abstract

Dental caries in the preschool population presents a significant challenge in the field of global public health, including Ecuador. Early detection of this disease is crucial for developing effective strategies for prevention and promotion of oral health, which can have a substantial impact on the quality of life of preschool-aged children. This study evaluated 600 children aged 3 to 5 years attending preschool education centers using the ICDAS II diagnostic criteria. The Student’s *t*-test was used to analyze differences between the means of two independent groups. Additionally, an analysis of variance (ANOVA) was conducted to assess differences between the means of three or more groups. The prevalence of caries was 87%, with a dft index of 3.85 in the three provinces studied. A high treatment need was observed in 84.17% of the population. No significant differences in the DFT index were found based on gender, although both groups exhibited elevated values. No significant differences were observed in relation to province and environment. The second molar was the most affected tooth, with a caries prevalence of 58.8%. Despite the lack of significant differences among the evaluated variables, a high prevalence and experience of dental caries were found in the studied population.

## 1. Introduction

Caries in preschool children has been on the rise in numerous countries and has emerged as a significant health concern, particularly among socially disadvantaged populations [1,2].

It is a multifactorial, sugar-induced disease that affects children under five years of age and causes the demineralization of hard tissue in primary teeth [3,4].

It is paramount to emphasize that dental caries in preschool-aged children is exceedingly common and demonstrates a high prevalence. About 1.76 billion children with deciduous dentition have been affected by this disease [5]. Based on an analysis of several research studies published by the United Nations between 2007 and 2017, it was found that the average prevalence of preschool caries is 23.8% in children under 3 years of age, and 57.3% in children between 3 and 6 years of age [5,6]. These results were confirmed by a systematic review of 80,405 children that showed 46.2% had a caries experience in their primary teeth. In addition, another systematic review using World Health Organization (WHO) criteria a found an overall prevalence of early invasive caries of 48% [5,7].

Individual factors associated with prevalence in children include the microbial composition of biofilms, the amount of sucrose and refined carbohydrates in the diet, salivary flow, tooth morphology, climate, fluoride in drinking water, toothpaste, and immunity to Streptococcus mutans [1]. Epidemiological studies have documented the association of early childhood caries (ECC) with factors such as low socioeconomic status, belonging to minority groups, low birth weight, and maternal transmission of microorganisms to the child. Additionally, certain behaviors related to pediatric oral health care, feeding practices, and hygiene have been found to be associated with ECC prevalence [8,9,10]. These include nocturnal bottle feeding, frequent consumption of cariogenic foods, delayed initiation of toothbrushing in children, and irregular brushing habits. It is important to highlight that today, it is common for young children and preschoolers to be under the care of caregivers other than their parents, such as grandparents, nannies, and childcare centers. This dynamic can influence the establishment and effects of poor habits associated with the development of caries in preschool children [8].

Another important factor to consider is the residential environment. International studies have yielded divergent results regarding childhood dental health in urban and rural areas, especially in developing countries [9,10,11,12,13]. In both developed and developing countries, rural areas are characterized by two key factors: limited availability of dentists and higher levels of poverty. These factors hinder children’s access to dental healthcare services, thereby increasing the risk of dental caries [14]. In summary, the provided context highlights the significance of the environment in the development of dental caries in children, particularly in relation to the availability of dental healthcare services, poverty levels, access to water fluoridation, and other factors associated with country development and urbanization [15].

In general, the consequences of preschool caries generate new carious lesions that affect not only the primary dentition, but also the permanent dentition [16,17]. In addition, high demand for emergency consultations, hospitalizations, high treatment costs, and absence from school ensue; in other words, preschool caries directly affects the masticatory and phonatory functions, social interactions, and cognitive development of affected children, with a consequent decrease in the quality of life, not only in the children, but also their parents, who suffer from the resultant stress as well [18,19].

Understanding the factors that contribute to poor health and the development of preschool caries is crucial for the development of appropriate and effective health policies. This knowledge enables us to identify the underlying causes of these oral health issues and design strategies that efficiently and preventively address this condition in the child population.

Malnutrition is a prominent health issue in Ecuador, as well as in other low- and middle-income countries [20,21]. A global phenomenon known as “nutritional transition” has been observed, wherein traditional diets are being substituted by low-quality foods and beverages high in sugar, fats, and carbohydrates, while lacking in essential micronutrients [22,23].The latest nutrition survey in the country reported that approximately 24% of children under the age of five experience this problem, with a tendency to become chronic in 6.7% of cases [15]. Parallel to the increase in childhood malnutrition, there has been a rise in the prevalence of chronic diseases such as obesity. Regarding oral diseases, dental caries in children has shown high levels of prevalence and experience [24,25]. Although studies focusing specifically on preschool-aged children are limited in the country, a strong association has been found between low maternal education and a higher prevalence and severity of dental caries in children from rural indigenous communities [26,27,28]. 

This study focuses on the southern region of Ecuador, encompassing both urban and rural areas in the provinces of Cañar, Azuay, and Morona Santiago. This region is characterized by a significant migrant population, and like the rest of the country, dietary habits have led to increased rates of malnutrition [21,23]. Given these circumstances, there is a recognized need to conduct this pioneering study in these regions, with the aim of gathering information on the prevalence and experience of dental caries in preschool children. The International Caries Detection and Assessment System (ICDAS II), a widely accepted and utilized method for identifying and classifying carious lesions, will be employed for this purpose [29].

Our hypothesis suggests that there is a substantial presence of dental caries in the study population, and furthermore, we propose that there are no significant differences in the prevalence and experience of dental caries when considering gender and environment.

## 2. Materials and Methods

This study was observational and cross-sectional, and is composed of data collected during an epidemiological survey conducted in 2019 of students from schools in southern Ecuador. Our research complied with the ethical principles established in the Declaration of Helsinki and with data protection regulations. In addition, the Board of Directors approved this study through Resolution No.048-C.D-2019, issued on 14 February 2019. The parents or guardians of the children were duly informed about the study and gave consent for their children to participate in the study. 

The diagnosis of carious lesions was conducted using the ICDAS II criteria, with a cut-off point of scores between 3 and 6. This approach was selected based on the evidence supporting its utilization in studies involving preschool, school-aged, and adolescent children [30]. It has been demonstrated that this combination of criteria minimizes the discrepancies between the World Health Organization (WHO) standard and the ICDAS II criteria.

The examined variable was the prevalence of caries, which represents the proportion of preschool children affected by the condition and was categorized as = 0 without caries and > 0 with caries, ICDAS II 3-6/CG > 0.Index of decayed and filled primary teeth (dft) ICDAS II/dft-ICDAS II 3-6.The independent variables were gender, province (Azuay, Cañar, and Morona Santiago), and environment (urban/rural).

### 2.1. Sample

A sample size calculation was performed, taking into account a 99% confidence level and a 2.5% margin of error. The evaluation of the correlation of the sample obtained an effect size of 0.3 and found a statistical power of 99.9%, with a probability of error of 0.043. For this, data were taken from 670 students from 3 to 5 years of age, randomly selected, and proportionally stratified according to environment by province, with 48% of the children from urban areas and 52% from rural areas.

Subsequently, it was verified that the participants met the inclusion criteria and that there were no legal or systemic impediments, in addition to obtaining informed consent. Finally, a final sample of 600 data was formed from the rural and urban settings of the studied provinces.

### 2.2. Calibration

To ensure accuracy in data collection, the eight professionals in the field of dentistry investigators in charge were instructed in the diagnosis of dental caries using ICDAS II criteria and OMS indexes. The training was given by certified professionals in the field; carious teeth, extracted teeth, and clinical sessions were used for the training. As a result of this training, a concordance of 0.83 was achieved, evaluated using Kappa and Cohen’s test.

Furthermore, additional measures were taken to ensure the quality of the re-copied data, such as constant supervision by a team of experts in the field and regular review of the records. This ensured that the results obtained were accurate and reliable for further analysis.

### 2.3. Examination

The clinical examination was carried out in classrooms with natural and artificial lighting provided by the educational centers. Information was collected from the participants before the examination was carried out using forms previously designed to record relevant data such as age, gender, school institution, ethnicity, geographic location, and type of locality (urban or rural). The examiners applied parameters and regulations taken from the WHO [31], applying all the biosafety rules and verifying that the dental surfaces were clean, dry, and well illuminated.

For the clinical examination, various instruments and materials were used, including an OMS-type periodontal probe, a flat intraoral mirror No. 5, nitrile gloves, disposable surgical masks for each patient, a head lamp, and gauze and cotton to control humidity. It should be noted that the examiners who carried out the examination were previously trained and were accompanied by assistants who were responsible for completing the data collection form.

The information obtained from each child at the time was recorded on cards and subsequently stored in an Excel sheet, data from which are available with the authorization of the parents or responsible guardian; this database was subsequently analyzed with the coded cards to obtain the data and then the statistical analysis of the study.

### 2.4. Statistical Analysis

The data collected during the investigation were recorded on forms designed according to the OMS guidelines [31]. The obtained data were analyzed using measures of central tendency and dispersion, and presented as percentage frequencies. The Kolmogorov–Smirnov test (*p* > 0.05) was employed to assess the normality of the data. Statistical analyses were conducted to compare the means of the groups in the study. The Student’s *t*-test was utilized to analyze differences between the means of two independent groups. Additionally, an analysis of variance (ANOVA) was performed to evaluate differences between the means of three or more groups. A significance level of α = 0.05 was applied to determine whether the observed differences were statistically significant in both tests. Different statistical techniques were employed for data analysis using the SPSS V28 software. The statistical programs IBM^®^SPSS v.27 and JASP^®^0.16.2 were also utilized to ensure precision in the analysis. The level of statistical significance was set at *p* < 0.05. 

## 3. Results

In Ecuador, the territorial division is based on a system of 24 provinces, instead of states, which constitute the administrative structure of the country. Each province presents unique geographical, cultural, and economic particularities, covering both urban and rural areas, the latter of which are characterized by having a smaller population quantity in relation to urban areas.

### 3.1. Distribution of the Participants

This study focused on the evaluation of 600 preschool children from urban and rural settings of the provinces of Cañar, Morona Santiago, and Azuay. Proportional stratification was applied while selecting participants, ensuring that both contexts were represented in similar proportions (Table 1).

### 3.2. Prevalence of Dental Caries in the General Study Population

The total percentage of carious lesions is reported in the results, exceeding 87% for primary dentition; healthy teeth comprised 13% of the sample (Figure 1).

### 3.3. Indicator of Caries in Children Aged 3 to 5 Years

In the examined group, a dental caries index of 3.85 was recorded, indicating the presence of a moderate level of caries in the evaluated population. Additionally, a sound teeth index of 5.56 was observed, while the decayed teeth index was 3.55.

The indicator of filled teeth was very low, at only 0.29. The treatment need persisted in 84.17% of the participants (Table 2).

### 3.4. Experience of Caries in Children According to Gender

No significant differences were found between either gender in relation to the dental caries index. However, elevated and similar values were observed regarding the treatment need in both groups. On one hand, it was found that boys had a slightly higher index of intact primary teeth compared to children. The treatment need was lower in girls, with 83.22% compared to 85.10% in boys, although this difference was not statistically significant (*p* > 0.05). Boys and girls demonstrated a comparable average rate in terms of treatment necessity (Table 3).

### 3.5. Indicators of Primary Dentition According to Caries Experience by Environment

A significant difference was observed in the index of healthy teeth between the urban and rural settings. In the rural areas, the average number of healthy teeth was 5.8, equivalent to 6 teeth in the oral cavity, while in the urban areas the average was 5.3, equivalent to 5 healthy teeth. Regarding the DFT index, both environments presented similar average values and no significant difference was observed between them. The DFT indices of the two environments were all at an average level and there was no significant difference. The need for treatment was 84% in both groups (Table 4).

### 3.6. Prevalence of Caries by Province

The average numbers of dental caries in the provinces of Azuay and Cañar remained stable and no significant differences were observed. However, in the province of Morona Santiago, there was a tendency to present a higher average compared to the other two provinces evaluated (Table 5).

### 3.7. Frequency of Caries According to Each Primary Tooth

More than half of the participants had caries in the upper and lower primary teeth, with greater involvement in some specific teeth. For example, 52.1% of the participants had caries in the upper lateral incisor, while 58.8% had caries in the upper second molar.

In the lower arch in the deciduous dentition, the central incisor was the most affected, with a caries rate of 53.3%, followed by the primary first molar, with a rate of 53.7%. These findings are shown in a graph that clearly visualizes the distribution of caries in the different teeth evaluated (Figure 2).

## 4. Discussion

In this initial study conducted in the country, an analysis was carried out using a representative sample of preschool children, revealing a high prevalence of dental caries according to the diagnostic criteria of ICDAS II. This finding is of significant importance as it highlights the magnitude of the caries problem in this specific population.

According to the findings of this study, a significantly high incidence of dental caries has been observed in the evaluated pediatric population, with no statistically significant differences found in relation to gender or place of residence. These results are consistent with previous research conducted at the national level in Ecuador [32,33], which, in examining the policies and interventions implemented by the country’s authorities, have highlighted the ongoing health challenges, particularly among the vulnerable population characterized by economic limitations and educational deficits. Although previous studies utilized the criteria proposed by the WHO [32,34], which do not consider the early stages of the disease, it is revealed how the presence and severity of carious lesions increase with the participants’ age.

However, more recent studies utilizing the ICDAS criteria [24,25,35] have consistently demonstrated that the geographical location of one’s residence has a significant impact on health indicators. Specifically, it has been observed that rural areas face specific challenges in terms of accessing healthcare services and having an adequate availability of specialized medical personnel [24]. These geographical disparities greatly influence the quality of medical care received and, ultimately, the health outcomes of the population [11].

The reported results agree with other studies carried out in other continents [10,36,37] where the high prevalence and experience of dental caries in primary dentition have been associated with both individual and environmental risk factors, such as high sugar consumption, malnutrition, and residential environment. Furthermore, other studies have identified dental caries in preschool age as an early marker of social disadvantage [38].

It is important to emphasize that regardless of the materials and techniques used to control the progression of lesions in these teeth, the actions may be inappropriate and insufficient [37,39,40] if they are not accompanied by supervised oral hygiene practices and educational processes from an early age [41,42], during which parents should raise awareness of the importance of keeping temporary teeth healthy.

Despite the great advances in the understanding of this disease, the results show how it continues to be very prevalent, relating its presence to the economic and social situation in countries [26,43]. The understanding of the disease has given rise to new prevention and treatment strategies [18]; however, it is still necessary to break down basic paradigms of care and approach, especially when it comes to permanent teeth, such as tooth 6, whose eruption occurs at an early age, in the presence of primary teeth prevalent in the mouth. Another point to analyze is the economic, educational, and social difficulties of the evaluated populations [44], which invite us to consider the need to establish education and health policies that meet the specific needs of these population groups.

The presence of carious injuries in early childhood is the main indicator of the socioeconomic level of the population [43] since it affects the general health of the individual, hindering the intake of food and the absorption of its nutrients [45], unleashing irreversible damage in the development of the child, as the dental position due to loss of structure and incorrectly executed dental restorations can affect their self-esteem [18]. Furthermore, a dangerous increase in the severity of the lesion, as seen in the study, proportional to the age of the individual, was linked to the lack of effective existing preventive measures and the excessive increase in the consumption and frequency of certain foods [44].

The lack of hygiene is another interesting factor that can explain the results of the study, and is related to the education and interest of parents in the health–disease processes that affect their children [3,5]. The transmission of habits and customs generally dependent on the mother is decisive in the presence of pathology at the first permanent molar [46,47]. Faced with this, multidisciplinary efforts aimed at educating the population are crucial, especially when it is understood that the presence of the lesions may interfere with the activities of patients and parents and/or caregivers [48], increasing the economic costs of treatment often associated with the use of full sedation due to the difficulty of handling existing lesions, which, in many cases [49], affect the economy and the quality of life of children and their families [50].

Current preventive measures focus on the use of fluoride in drinking water, oral hygiene utensils, and sealants in pits and fissures [51,52] that are massively applied in communities. Additionally, schools and health care posts offer dental care to parents and their children; however, these measures need to be redirected to children at an earlier age, or even to parents before their children are born. Our results reveal that it is necessary to review the control strategies implemented in this population regarding the consumption of sugars by using control measures and promoting individual activities focused on education in health processes and the importance of caring for temporary teeth from early childhood [47,53,54].

One of the existing limitations of this study is related to its descriptive nature, since the health status of the participants was not monitored, which made it impossible to truly evaluate the preventive measures adopted; therefore, it is necessary to structure future studies to evaluate the activities and policies proposed by the existing health organizations and ensure their continuity.

As dentists, we face the daily challenge of treating patients with malocclusion, dental absences, and even joint diseases [5]; these situations can be prevented with concrete actions that take age, risk, and family involvement into consideration [55]. This can help control the disease from its early stages, taking advantage of the fact that caries is a chronic, slow, and progressive disease [56] that requires support and education for mothers from pregnancy through the early post-natal period.

## 5. Conclusions

This first study in the country with a representative sample in preschool children found a high prevalence of caries under the ICDAS II criteria. Regarding gender, no significant differences were observed, although a slightly higher percentage occurred in boys than in girls.

Regarding the environment, both in rural and urban areas, there was evidence of a high need for treatment, and when analyzing by province, we found that Azuay and Cañar presented similar patterns, while Morona Santiago showed a higher percentage. According to the data analyzed in the study, dental caries was more frequent in the second molar, with a prevalence of 58.5%.

## Figures and Tables

**Figure 1 children-10-01123-f001:**
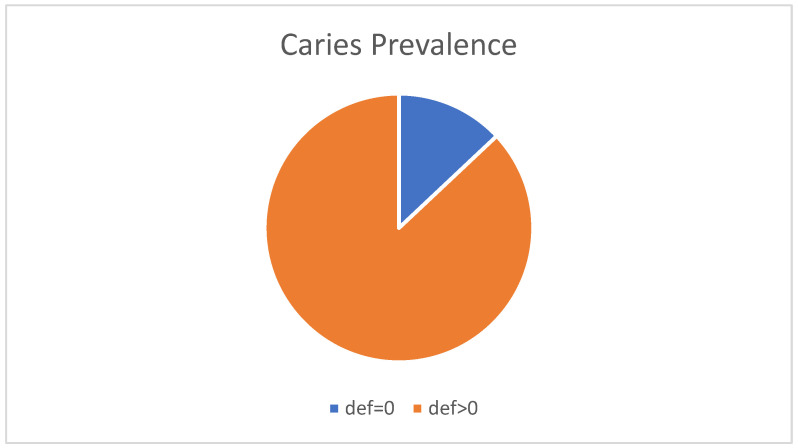
Caries prevalence in the total study population.

**Figure 2 children-10-01123-f002:**
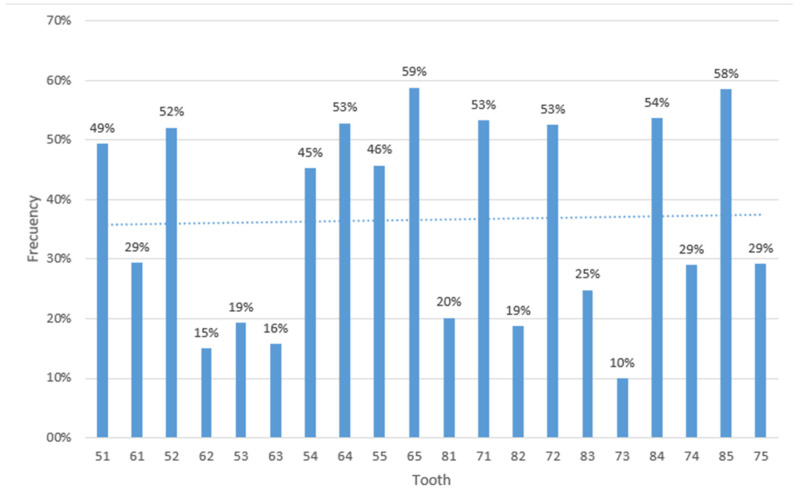
Distribution of caries according to the evaluated teeth.

**Table 1 children-10-01123-t001:** Distribution of the participants.

	Cañar	Azuay	Morona Santiago	Total
Urban	95	110	104	309
Rural	98	97	96	291
Total	193	207	200	600

**Table 2 children-10-01123-t002:** Caries experience in early dentition (DFT) in the study Region.

	Healthy	Carious	Obturated	Need for Treatment	dft
Media	5.56	3.55	0.29	84.17%	3.85
* SD	4.55	3.26	0.65	30.15%	3.38

Note: * SD: standard deviation. dft: index of primary dentition of decayed and filled teeth.

**Table 3 children-10-01123-t003:** Indicators of caries experience in primary dentition according to gender.

	Healthy		Decayed		Filled		Treatment Need		dft	
	M	F	M	F	M	F	M	F	M	F
Media	5.48	5.65	3.54	3.57	0.27	0.32	85.10%	83.22%	3.81	3.89
SD	4.65	4.45	3.29	3.24	0.59	0.71	29.19	31.10	3.39	3.37
T	−0.821		−0.217		−1.643		1.259		−0.528	
* *p*	0.412		0.829		0.101		0.208		0.598	

Note: * *p* < 0.05 Significant difference. SD: standard deviation. T: T student. M: Male. F: Female.

**Table 4 children-10-01123-t004:** Indicators of primary dentition according to caries experience by environment.

	Healthy		Carious		Obturated		Need for Treatment		dft	
	Urban	Rural	Urban	Rural	Urban	Rural	Urban	Rural	Urban	Rural
Media	5.30	5.80	3.61	3.50	0.28	0.30	84.21%	84.15%	3.89	3.80
SD	4.32	4.74	3.33	3.20	0.60	0.70	30.13%	30.19%	3.42	3.34
T	−2.390		0.795		−0.851		0.040		0.602	
*p*	0.017 *		0.427		0.395		0.968		0.547	

Note: * *p* < 0.05 Significant difference. SD: standard deviation. T: T student.

**Table 5 children-10-01123-t005:** Indicators of primary dentition according to province by caries experience.

	Healthy			Carious			Obturated			Need for Treatment			dft		
	A	C	MS	A	C	MS	A	C	MS	A%	C%	MS%	A	C	MS
M	5.63	5.24	5.93	3.46	3.52	3.73	0.29	0.25	0.35	84.08	83.32	85.40	3.75	3.77	4.08
SD	4.33	4.84	4.36	2.98	3.59	3.12	0.60	0.58	0.80	30.18	31.11	28.83	3.09	3.68	3.28
F	3.677			1.071			3.747			0.628			1.723		
*p*	0.025 *			0.343			0.024 *			0.534			0.179		

Note: A: Azuay, C: Cañar, MS: Morona Santiago. Note: * *p* < 0.05 Significant difference. SD: standard deviation. F: analysis of variance (ANOVA).

## Data Availability

https://ucacueedu-my.sharepoint.com/:x:/g/personal/mvelezl_ucacue_edu_ec/Ecu70alhu7pEmk-HB8fAcZABOC1qb-AMfh48ijYgmfQIYA?e=3NC5vh (accessed on 2 February 2023).

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
