# Peer review of "Caries Experience in Preschoolers in Three Ecuadorian Communities"

_children, 2023, doi:10.3390/children10071123_

Round 1

Reviewer 1 Report

This manuscript has a subject that may interest readers. However, making the following minor corrections will increase the value of the article.

Abstract: Well written.

Introduction:

·      The statement "ECC affects 1.76 billion children" in lines 36-37 is contradictory. This is a referenced data from article 4, Zou et al. However, although this data is available in the article of Zou et al., they also referred it from the article of Vos et al. When we reached the base article, we could not find any statement about this data. Therefore, it would be more appropriate to remove the relevant expression from the manuscript.

·      The sentence on lines 42-44 has no reference. Please add reference.

Materials and methods: What are the number of personnel calibrated in the calibration header mentioned in lines 97-106? What is the medical experience of the relevant personnel? Please give details about this.

Results: The data presented on the map about Ecuador and its states can be informative for those who have no idea about the country.

Discussion:

·      In this section, it was mentioned that similar findings were obtained with other studies, but no details were given. Comparisons with studies in the literature should be given more space.

·      This section contains a lot of judgment. Care should be taken to make comments based on evidence rather than judgments.

·      In addition, it may be useful to give some general information about the states mentioned in the study in general.

There is no noticeable mistake in terms of language.

Author Response

thank you for your comments. We have followed your recommendations and we have attached a document with the dellate.

Reviewer 2 Report

The paper Caries Experience in Preschools in Three Ecuadorian Communities describes caries lesions in children aged between 3-5 years, however the ethical approval is refering to chidren aged 6,12,and 15 years. How do the authors explain this discrepacy?

Explain every abbreviation used in the abstract and manuscript, before the abbreviation is used. 

What is the originality of the study?

Some references are very old.

Author Response

thank you for your comments. We have followed your recommendations and have attached a document with the description.

Reviewer 3 Report

Dear authors,

Your manuscript entitled CARIES EXPERIENCE IN PRESCHOOLERS IN THREE ECUADORIAN COMMUNITIES refers to an important topic and contains useful data.

However, in my opinion, there are some important aspects that should be addressed. Please find below some recommendations for improvement.

The Title section:

-differences exist between the full text article and the Abstract presented

The Abstract section:

-line 15 - please consider "preschoolers" or "preschool children" instead of "preschooler children"

-line 23 - "0,05"- please pay attention to the expression of numbers in English

-lines 22-23 - please reformulate for a better understanding

-please consider "gender" instead of "sex" throughout the manuscript

-lines 25-26 - "By setting, the COD was found to be within a moderate level, with no significant differences. " - please reformulate

-line 26 - "58.8% caries incidence" - was incidence assessed in the study?

The Introduction section:

-line 35 - ECC is the abbreviation of Early Childhood Caries

-line 40 - please pay attention to references citation, and throughout the entire manuscript as well

-more information are needed in this section concerning childhood caries, the characteristics of the studied provinces (geographical, demographical, social)

-please include test/null hypothesis in the end of this section

The Materials and methods section:

-why did the authors use COD index? Were permanent teeth assessed, too? In this case, separate data should be found in the Results section

-line 99 - what does "ICDAS 3 criteria" mean?

-please specify how many investigators were involved in the study

-line 100 - "images" - what kind of images do you refer to? Were X-Rays taken?

-line 109 - "academic grade" - whose? of the parents? Please specify

-where did the clinical examination take place? In dental settings? Please specify

-were any X-Rays taken?

The Results section:

-Figure 1 - please consider replacing "Healthy/sick" in the legend with "def=0/def>0", or "caries-free/caries-affected"

-subsection 3.3:

-please consider replacing "Prevalence" in the title with "Caries indicators"

-please explain what "rate of healthy teeth" means - how was it calculated? 

-please explain what "rate of decayed teeth" means - is it the percentage of dt in the entire dft?

-Table 2 - please consider changing the expression "ECC", and throughout the entire manuscript as well. ECC refers to a specific pattern of the disease, not merely to dental caries. Also, please add the symbol % to percentage values, and explain what is "DE". On the other hand, "SD" is explained under the Table, but does not appear inside it. Also, "dft" is not a rate (it should be expressed as a percentage in this case), but the value of the index. Also, the symbols "*" are not found inside the Table, but only under it

-subsection 3.4:

-line 158 - please explain what does "preexistence" of caries mean?

-Table 3 - it has to be corrected, as it is incomplete. Also, please explain what abbreviations mean (DE, T, P)

-subsection 3.5:

-please consider changing the title - "Incidence" is a different indicator, which cannot be assessed in cross-sectional studies

-please explain what "COD" index means

-Table 4 - please add % to percentage values. The same observations as for the other Tables

-"p" as cut-off point of statistical significance should be written with lowercase letter, italicized

-subsection 3.6:

-Table 5 - the same observations as for the other Tables

-Figure 2 is confusing - please consider to express only % carious

The Discussion section:

-lines 197-200 - the statement is not justified by the methods used and the obtained results

-lines 205-206 - how could the results of the present study explain social and economic disparities? This aspect was not studied in this research. If the differences obtained by province could explain this, please specify in what way

-lines 212-213 - "caries incidence" was not assessed in this study

-why were correlation coefficients mentioned in the "Materials and methods" section? No results are presented concerning correlations

-line 224 - "permanent teeth" were assessed in the present study?

-line 227 - not the "descriptive nature", but the cross-sectional design of the study

-please add comparisons with the results of similar studies in the literature

The Conclusions section:

-line 241 - "incidence" was not assessed in the present study.

Minor editing is required

Author Response

(The authors gave the same response as above.)

Round 2

Reviewer 2 Report

There is still a discrepancy:

The paper Caries Experience in Preschools in Three Ecuadorian Communities describes caries lesions in children aged between 3-5 years, however, the ethical approval is referring to children aged 6,12, and 15 years. How do the authors explain this discrepancy?

Author Response

Dear Reviewer,

 We inform you that to inform you that the scope of the project has been promptly expanded to include preschoolers. We have attached the relevant document for your review. We greatly appreciate your attention to this matter and eagerly await your valuable feedback.

Sincerely,

The autors

Reviewer 3 Report

Dear Authors,

Thank you for considering most of my recommendations concerning the first version of your paper entitled CARIES EXPERIENCE IN PRESCHOOLERS IN THREE ECUADORIAN COMMUNITIES.

However, some points still need to be addressed.

In the Introduction section:

-the research is not about Early Childhood Caries (ECC). As I mentioned in my previous report, this is a special pattern of dental caries. Please remove from the present paper all the information concerning ECC, as it was not assessed.

-line 131 - please replace "sex" with "gender"

In the Results section:

-line 258 - please rephrase so that the sentence does not start with "While"

-line 272 - please replace "found between both gender" with "found by gender"

-Table 3 - it is incomplete, or maybe the rows are not all visible: what do the values in the "Media" row, expressed by gender, represent?

In the Discussion section:

-line 346 - please replace "sex" with "gender"

-please add more results from similar studies

In the Conclusion section:

-please remove the final paragraph

Minor editing of English language is required.

Author Response

I sincerely appreciate your valuable comments and suggestions on my article. Your review has been greatly helpful in enhancing the quality of my work.Attached is a document detailing the corrections.
